# Use of Lactic Acid Bacteria During Pikeperch (*Sander lucioperca*) Larval Rearing

**DOI:** 10.3390/microorganisms8020238

**Published:** 2020-02-11

**Authors:** Carlos Yanes-Roca, Eric Leclercq, Lukas Vesely, Oleksandr Malinovskyi, Tomas Policar

**Affiliations:** 1South Bohemian Research Center of Aquaculture and Biodiversity of Hydrocenoses, Faculty of Fisheries and Protection of Waters, University of South Bohemia in Ceske Budejovice, Zátiší 728, 389 25 Vodňany, Czech Republic; veselyl@frov.jcu.cz (L.V.); omalinovskyi@frov.jcu.cz (O.M.); policar@frov.jcu.cz (T.P.); 2Lallemand SAS, 19 rue des Briquetiers, 31702 Blagnac, France; eleclercq@lallemand.com

**Keywords:** lactic acid bacteria, live feed, first feeding, *Sander lucioperca* larvae

## Abstract

This trial tested the use of lactic acid bacteria (LAB) on pikeperch (*Sander lucioperca*) larvae during their first feeding. The trial included the use of two probiotic treatments and one control (no probiotics). Pikeperch larvae were exposed to LAB as follows: (1) the live feed (Treatment 1, live feed) or (2) via the live feed and the larval rearing water (Treatment 2, probiotic). Significant differences were found between the treatments in terms of total length (TL), myomere height (MH), overall survival, and the tolerance to a high salinity challenge. Larvae exposed to LAB via both the live feed and the rearing water had a significantly higher overall survival rate (85%) than the other two treatments at 21 dph. When both treatments were subjected to high salinity rates (18 parts per thousand (ppt)), both treatments exposed to LAB demonstrated higher survival rates than the control treatment (28% and 40% survival rate at 180 min for the live feed and probiotic treatments, respectively, as compared with a 100% mortality rate at 150 min for the control). At the same time, larvae exposed to the probiotic treatment had a significantly higher TL as compared to the control after 12 and 21 days post hatch (dph) (probiotic 7.13 ± 0.21 and 11.71 ± 1.1 mm, control 5.86 and 10.79 mm at 12 and 21 dph, respectively). The results suggest that the use of LAB in both the live feed and the rearing water has a positive effect on pikeperch larval quality by strengthening their resilience to stress conditions, as well as improving the growth and survival rates.

## 1. Introduction

Pikeperch (*Sander lucioperca*), a fresh and brackish water fish belonging to the *Percidae* family, is in high demand by recreational anglers and the gastronomic industry [1,2]. Because of the high demand, pikeperch is currently one of the targeted species included in the European Union’s plans to diversify the inland freshwater aquaculture. However, larviculture development in recirculating aquaculture systems (RAS) are encountering several obstacles such as low stress resistance, nutritional deficiencies, and cannibalism which result in low survival rates during the larval stage [3]. In order to supply the larvae with adequate nutrition, live feed is required.

Recently, rotifers were introduced to first feeding of pikeperch larvae with successful results [4,5,6]. Rotifers have the ability to absorb and retain the nutritional composition of any given diet that it is exposed, a quality that supplies pikeperch larvae with an adequate prey size and optimal nutrition [7]. These nutrients include highly polyunsaturated fatty acids that are essential for the survival of pikeperch [8,9,10]. However, the use of live feeds during a first feeding also introduces pathogenic bacteria into the closed system [11].

Probiotics are “live microbial feed supplements which beneficially affects the host animal by improving its intestinal microbial balance” [12]. Exposing fish larvae to selected probiotics has been proven to improve their health and increase their resilience to pathogens and disease due to the gastrointestinal microbiota dependency on the external environment [13]. Probiotics also compete with pathogens for nutrients and adhesion sites, which help to stimulate the immune system [13]. It is also important that defense mechanisms are present in the immune system before it is fully developed, so that non-pathogenic bacteria proliferate and inhibit colonization [14]. Probiotics play a role by occupying receptor sites and competing for food, thus, preventing detrimental bacteria in fish larvae from colonizing [14]. Additional benefits of probiotic bacteria supplementation through live feeds include improving the nutritional and growth performance of larval fish reared in RAS [15]. The bacteria contribute to the digestion of dietary macromolecules in the developing larval gut digestive system [16]. Moreover, recent studies with pikeperch [17,18] have also documented the beneficial effects of adding probiotics to the diet.

Most probiotic microorganisms belong to lactic acid bacteria (LAB) [19]. LAB are gram-positive, usually non-motile, nonsporulating bacteria that produce lactic acid as a major or sole product of fermentative metabolism. Nutritionally, LAB are fastidious, requiring carbohydrates, amino acids, peptides, nucleic acid derivate, and vitamins. The fish larvae’s gut is sterile until hatching, but soon after hatching, it comes in contact with the environment and live feeds lead to successive colonization by a variety of microbes [20,21]. The balance of this microbiota is influenced by a variety of factors including, but not limited to, feed type, animal physiology, and immunological factors. Such microbiota in endothermic animals are dominated by gram-positive bacteria such as LAB [22,23]. LAB are characterized by the following: (1) cell-surface properties for mediating adhesion, (2) survival within the gastrointestinal tract, (3) resilience to stress conditions, (4) ability to produce antioxidants, (5) antimicrobial effects, and (6) the positive influence on the immune system [24].

### Objective

The aim of this study was to evaluate the influence of *Pediococcus acidilactici* MA 18/5M on pikeperch larval rearing during the first 21 days post hatching (dph).

## 2. Materials and Methods

The trial was run at the University of South Bohemia, Facility of Fisheries and Protection of Waters, Czech Republic (USB, FFPW). Spawning and fertilized egg production was from pond-cultured pikeperch broodstock [25,26] (TL = 517 ± 35 mm and W = 1215 ± 200 g) held at the same facility under controlled conditions [27] in RAS. Final oocyte and sperm maturation were performed under a 15 h:9 h light/darkness regime with a light intensity of 100 lux, and a water temperature of 15 ± 0.5 °C [27,28,29]. It was synchronized with an intramuscular hormonal injection of 500 IU·kg^−1^ of Human Chorionic Gonadotropin (hCG; Chorulon, Intervet International B.V. Ljubljana, Slovenia), as previously done by Křištan [30] and Blecha [28]. All broodstock were anesthetized with clove oil (Dr. Kulich Pharma Ltd., Hradec Králové, Czech Republic) at a concentration of 30 mgL^−1^ [31] before manipulation. After hormonal treatment, pairs of both sexes were separated and stocked in RAS tanks for nest spawning, as previously studied by Malinovskyi [25,26]. After spawning, egg fertilization, and laying, broodstock were removed and eggs on the nest were incubated in each tank under a water temperature of 16 ± 0.5 °C, for 8 days until hatching occurred [25]. Three-day old larvae were stocked at 100 larvae per liter into 2 L larval rearing tanks (*n* = 12). Water quality parameters, salinity (3 ± 0.5 ppt), dissolved oxygen (8.0 ± 1 mgl^−l^), temperature (17.1 ± 0.2 °C) in the RAS were monitored daily. Ammonia (NH_3_ = 0.20 ± 0.05 mgL^−1^)_,_ nitrite (NO_2_ = 0.02 ± 0.01 mgL^−1^) and nitrate (NO_3_ = 0.10 ± 0.03 mgL^−1^) levels were measured every 3 days.

Three treatments were tested in quadruplicate. The first was the control treatment, where larvae were offered rotifers fed with *Nannochloropsis occulata* for the first 11 days of exogenous feeding (15 dph) followed by unenriched artemia until the end of the trial (21 dph). No probiotics were used during this treatment. The second treatment (live feed) used the same live feed protocol with the addition of the probiotic (Bactocell Aqua 100, *Pediococcus acidilactici*, 1.10^11^ CFU/g of product, Lallemand SAS, Blagnac, France) at a daily dose of 1 g/m^3^ in the rotifer and artemia culture tanks, giving a probiotic concentration of 1.10^5^ CFU/mL in the live feed culture water. In this treatment, the probiotic was only used on the larval feed (rotifer and artemia), therefore, larvae had no direct external contact with the probiotic. Their contact with the probiotics was limited to only when ingesting the prey. The third treatment (probiotic) followed the same live feed protocol as the control treatment with rotifers and artemia. They were exposed to a daily dose 1 g/m^3^ of Bactocell Aqua 100 during their culture. Additionally, the probiotic product was added daily to the larval rearing water at a dose of 0.1 g/m^3^ daily over the trial’s duration. In this treatment, larvae had direct external contact with probiotics (in the water), as well as when they were ingesting prey.

Rotifers were fed to the larvae three times per day (08:00, 11:30, and 15:30) starting at 4 days post hatching (dph) until 15 dph, with an initial concentration of 10 individuals per ml. Artemia were fed to each experimental group from day 12 post hatching. Feeding densities were steadily increased based on residual counts, performed prior to each feeding (Table 1). By 21 dph, rotifer density was 0 rotifers ml^-1^ and 8 artemia ml^−1^.

Live feed culture for the trial was done onsite. Rotifers (average size of 280 µm) were produced following a batch culture protocol fed with *N. occulata* (Nanno 3600, Reed Mariculture, Campbell, USA) at a rate of 1 mL of paste per liter of culture twice a day. Artemia nauplii’s average size was 430 µm. Flow rates started at 100 mL.min^−1^ and increased with time (Table 1). Prior to each feeding, flow was stopped and re-started two hours after, in order to improve larval feeding efficiency.

Eight and 12 days after treatment initiation (12 and 16 dph), 40 larvae per treatment (10 per tank) were collected using a 300 micron diameter mesh, and their total length (TL), myomere height (MH), eye diameter (ED), stomach fullness (SF), and air bladder inflation were recorded according to Yanes-Roca [4]. Recordings were made using an Olympus BX41 microscope fitted with a Canon-72 digital camera (Tokyo, Japan) and the Olympus (Tokyo, Japan) cellSens imaging software (version 1.3).

Prior to the appearance of cannibalism and light photosensitivity, the trial was terminated at 21 dph. At the end of the trial 21 dph, final survival was assessed. One hundred larvae per treatment (25 per tank) were assessed for morphometric analysis (TL, MH, ED, SF), and 100 larvae per treatment were collected and used for a subsequent salinity stress challenge.

### 2.1. Salinity Stress Challenge

Twenty-one days after hatching, 100 larvae per treatment (25 per tank) were collected and transferred to a 2 L tank (*n* = 3), where they were exposed to a salinity of 18 ppt for three hours. Larval mortality was recorded in each tank every 10 min during the first hour, then, recordings were taken at 120, 130, 140, 150, and 180 min from the initial stocking. Water quality conditions were kept the same as the original trial tanks, with the exception of salinity (18 ppt).

Larvae during this trial were handled in accordance with national and international guidelines for the protection of animal welfare (EU-harmonized Animal Welfare Act of the Czech Republic). The experimental unit is licensed (no. 2293/2015-MZE-17214 and no. 55187/2016-MZE-17214 in project NAZV QK1820354) according to the Czech National Directive (Law against Animal Cruelty, no. 246/1992).

### 2.2. Statistical Analysis

Differences between the body measurements and stomach fullness in three different treatments of larvae (sampled at 12, 16, and 21 dph) were evaluated with linear mixed models (LMM, package *lme4*, version 1.1-7; [32]). The effect of the different probiotic treatment was tested on fish TL, MH, and ED (response variables). The tank was included as a random effect. Prior to LMM, the different response variables were transformed with the Box-Cox transformation, which gives the best power estimate for each variable (package *car*, version 2.1.2; Fox and Weisberg, 2011; [33]). Thereafter, multiple pairwise comparisons between treatments were obtained using Tukey’s all-pair comparisons, applying the Bonferroni correction to adjust the *p*-values (package *multcomp*, version 1.3-3; [34]).

Differences in stomach fullness (1 to 4, 1 being an empty gut and 4 a full gut) were evaluated with generalized linear mixed models (GLMM, package *lme4*), fitted with a binomial error structure. Stomach fullness was used as response variable and the tank as a random factor. These analyses were followed by multiple pairwise comparisons with Tukey’s all-pair comparisons.

The pikeperch fish survival rate was compared between treatments using a generalized linear mixed model (GLMM), with the survival fish (i.e., proportion of alive fish at 21 dph as a response variable) fitted with a binomial error structure, and with enrichment as a fixed effect and the tank as a random effect. After GLMM, pairwise comparisons were obtained with Tukey’s all-pair comparison test. A Bonferroni correction was applied to adjust the *p*-values of multiple comparisons.

To test the salinity stress tolerance response among the treatments, a non-parametric survival analysis (Kaplan–Meier method) was performed for all groups, using survival package (Therneau and Grambsch, 2000).

## 3. Results

### 3.1. Larval Growth

At the start of the trial (3 dph, prior first feeding), pikeperch larval TL and BW was 5.25 ± 0.5 mm. At 12 dph, the probiotic treatment group (Figure 1) had the larvae with the largest average total length (7.13 ± 0.21 mm). By the end of the trial (21 dph), the average total length was significantly greater (LMM, *p*-value <0.05) in the probiotic treatment (11.71 ± 1.15 mm) than in the control and live feed treatments (Figure 1), but no significant treatment differences (LMM, *p*-value > 0.05) were found in total length at 16 dph.

When looking at myomere height (Figure 2), no significant differences were detected (LMM, *p*-value >0.05), with an exception at 16 dph, where a treatment effect was found (LMM *p*-value <0.001). When looking at the eye diameter, no significant differences were found (LMM, *p*-value >0.05, data not shown).

No significant differences in stomach fullness of larvae were found between treatments (GLMM *p*-value <0.05) and all prey were ingested by larvae, regardless of treatment (Figure 3).

### 3.2. Survival

Survival rates at 21 dph were significantly different between treatments (GLMM and pairwise comparisons *p* < 0.001), showing that the survival of larvae exposed to the probiotic treatment was 1.7 times higher than larvae from the control treatment and 1.53 times higher than larvae exposed to the live feed treatment, whereas the survival of larvae from the live feed treatment was 1.1 times higher than of larvae from the control treatment (not significant p > 0.05) (Figure 4).

### 3.3. Salinity Stress Tolerance

Larvae exposed to 18 ppt salinity from the different treatments reacted differently over time (Figure 5). On the one hand, larvae from the control treatment experienced mortality from the beginning of the exposure, having a 20% mortality after 30 min of exposure. On the other hand, larvae from both the probiotic and live feed treatment had a 13% mortality during the same period of time. After one hour of exposure, larvae from the control treatment had the highest mortality (52%), followed by the live feed treatment (48%), and the probiotic treatment (29%). Mortality in the control treatment slowed down during the following hour, resulting in an overall mortality rate of 70% (an 18% mortality rate increase). In contrast, tanks with larvae from the probiotic and live feed treatments experienced no mortality during the same period of time (Figure 5). During the following 60 min, a mortality increase was observed in all treatments. No larvae were alive after 150 min of exposure time in the control treatment, while, in comparison, tanks from the probiotic and live feed treatments had surviving larvae after 3 h of exposure. The probiotic treatment had the lowest final mortality rate (67%) as compared with the live feed treatment (75%); a significant difference (*p*-value <0.05) was found in the mortality rates between treatments after three hours.

## 4. Discussion

As with other marine species of similar economic value, such as the grey mullet (*Mugil cephalus*) [35], sole (*Solea solea*) [36,37], gilthead seabream (*Sparus aurata*) [38,39], and sea bass (*Dicentrarchus labrax*) [40], the introduction of rotifers *(Brachionus plicatilis)* to pikeperch larval culture [4] has improved the survival rate and overall fitness.

However, with this new development in pikeperch larval culture, harmful pathogens and diseases can become a setback in larval hatcheries. Live food, such as rotifers, have a high bacterial load which becomes a factor for bacterial contamination, and therefore introduces diseases into larval cultures. Proliferation of harmful bacteria, such as Vibrio, are common in intensive aquaculture settings, therefore, several studies have stated the key importance of controlling the bacterial load in live food to reduce the negative effects [20,41,42]. The use of probiotics during this stage has been proven to control bacterial load in live food, as well as in the rearing environment [43]. The positive influence of probiotics during this experiment was observed when looking at other parameters such as total length and myomere height. Although it was not fully expected to be an outcome from this trial, growth was improved when using probiotics. Such results have also been observed in other species such as sea bass larvae, Nile tilapia (*Oreochromis niloticus*), swordtail (*Xiphophorus Heller* or *X. maculatus*), and guppy (*Poecilia reticulate* or *P. sphenops*) [44,45,46,47]. It has been speculated that such improvements in growth could be related to a positive effect on appetite stimulation or by just improving digestibility. Gastro-colonization is enhanced by probiotic microorganisms when administered over a long period of time, due to the higher proliferation rate as compared with the expulsion rate [16].

Carnevali [48] stated that probiotics positive effect on fish larvae is also due to the increase in levels of an insulin-like growth factor (IGF), which is responsible for muscle growth in fish. Such results, as well as a decrease in a growth antagonist such as myostatin (mstn), were observed in zebra fish (*Danio rerio*), sea bass, sea bream, and sole [49,50,51,52,53]. No metabolic analyses of such IGF receptors and binding proteins on pikeperch larvae were performed for this trial, but future work on this topic is recommended.

One of pikeperch culture’s main obstacles is their low tolerance to stress conditions, such as the handling and alteration of the fish’s physical conditions [1]. Such stress conditions are difficult to avoid when intensive culture methods of pikeperch are being developed to achieve mass production. The use of probiotics has been tested in several species, with the intent of increasing stress tolerance. Reducing cortisol levels [49,54], while increasing glycogen and triglycerides reserves in fish larval livers [55], are one of the benefits observed in other species, as well as increased antioxidant enzymes [56]. Such influences on the fish larvae metabolism could explained the results obtained with pikeperch, where stress tolerance was significantly enhanced with the use of Bactocell in both live feed and direct water application.

By the end of the trial, significant improvements of survival rates were observed, especially between the probiotic and the control treatment. As discussed earlier, such results can be attributed to the many benefits that probiotics have with fish larvae. Another benefit not discussed yet is the potential influence that some probiotic strains (gram positive) have over water quality by transforming organic matter to CO_2_ [57]. This could have had an influence in the survival differences between the live feed and probiotic treatments. Although no significant differences in ammonia, nitrate, and nitrite were found between treatments, a slight pattern was observed. The probiotic treatment had a slightly lower level of ammonia (NH_3_ = 0.18 ± 0.03 mgL^−1^) and nitrate (NO_3_ = 0.08 ± 0.04 mgL^−1^) than the live feed and control treatment. Such differences could be due to the fact that larvae in the probiotic treatment were exposed to the probiotic through the live feed and the culture water, potentially giving higher quality water conditions than the live feed treatment, whose water was not exposed to Bactocell.

## 5. Conclusions

There was a positive correlation between the survival rate and fitness of pikeperch larvae when *Pediococcus acidilactici* MA 18/5M was used during the first 21 days post hatching. The use of such commercial probiotics during live feed cultures, as well as in the larval rearing water (probiotic treatment), significantly increased the survival rates and stress tolerance. The growth parameters also improved, indicating this LAB strain contributed to support pikeperch larval quality. This warrants further research on the benefits of probiotics on larval metabolism, digestive enzyme activities, and on controlling harmful bacteria which are all key parameters for sensitive species, such a pikeperch, to achieve optimal rearing efficiency.

More in depth research, specifically addressing metabolic effects and digestive enzyme activity, is also needed to help identify how probiotics can improve overall pikeperch growth.

## Figures and Tables

**Figure 1 microorganisms-08-00238-f001:**
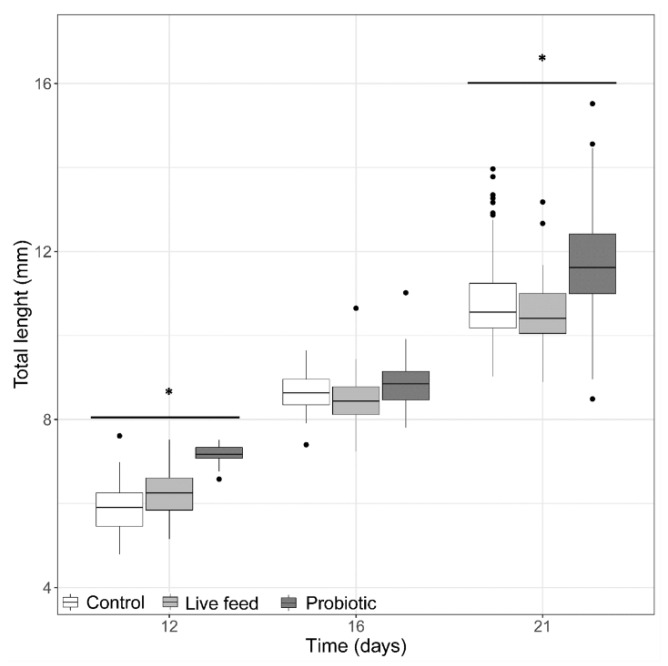
Larval total length from three treatments at days 12 (*n* = 40), 16 (*n* = 40), and 21 dph (*n* = 100). Dots shown are the out layers, whiskers indicate the maximum and minimum values excluding out layers, the line in the middle of box is the median value and upper and lower quartiles are the ends of the box. Statistically significant differences between treatments are marked with an asterisk.

**Figure 2 microorganisms-08-00238-f002:**
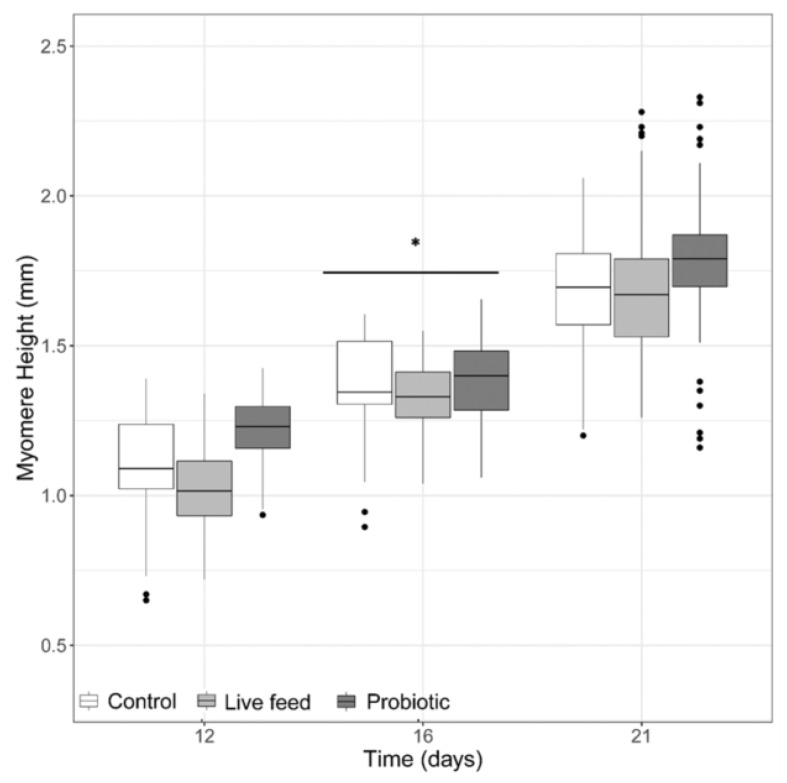
Larval myomere height from three treatments at days 12 (*n* = 40), 16 (*n* = 40), and 21 dph (*n* = 100). Dots shown are the out layers, whiskers indicate the maximum and minimum values excluding out layers, the line in the middle of box is the median value and upper and lower quartiles are the ends of the box. Statistically significant differences between treatments are marked with an asterisk.

**Figure 3 microorganisms-08-00238-f003:**
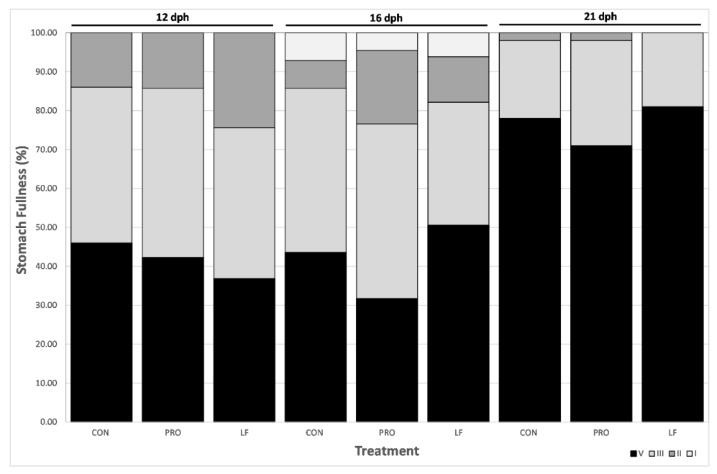
Larval stomach fullness from three treatments at days 12 (*n* = 40), 16 (*n* = 40), and 21 dph (*n* = 100). Expressed in percentage (1 to 4, 4 being the maximum fullness, from darkest to lightest grey).

**Figure 4 microorganisms-08-00238-f004:**
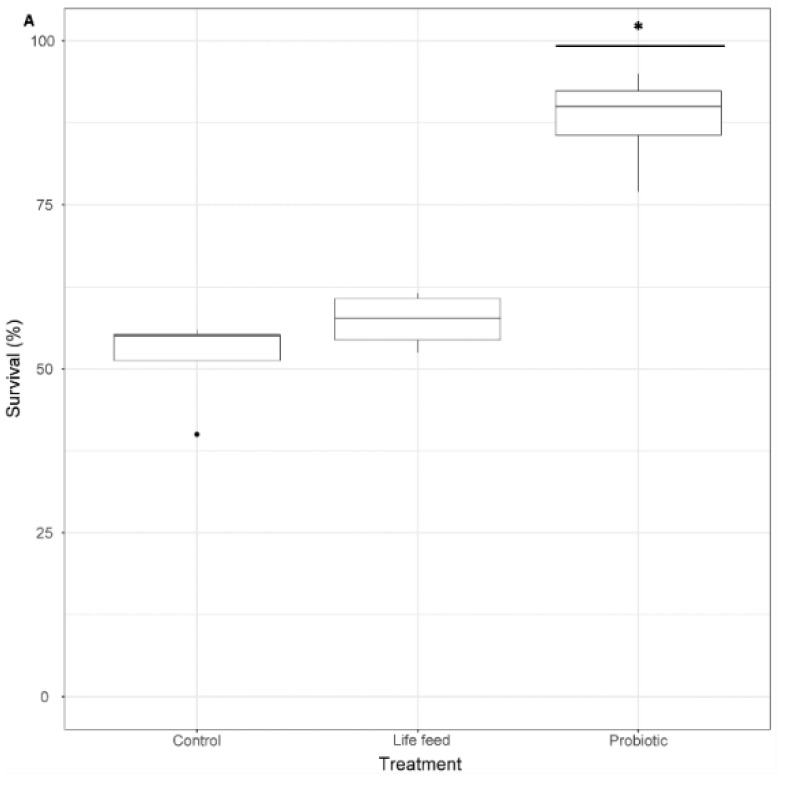
A: Larval survival percentage (*n* = 100), after 21 dph. Dots shown are the out layers, whiskers indicate the maximum and minimum values excluding out layers, the line in the middle of box is the median value and upper and lower quartiles are the ends of the box. Statistically significant differences between treatments are marked with an asterisk. Statistically significant differences between samples are marked with an asterisk.

**Figure 5 microorganisms-08-00238-f005:**
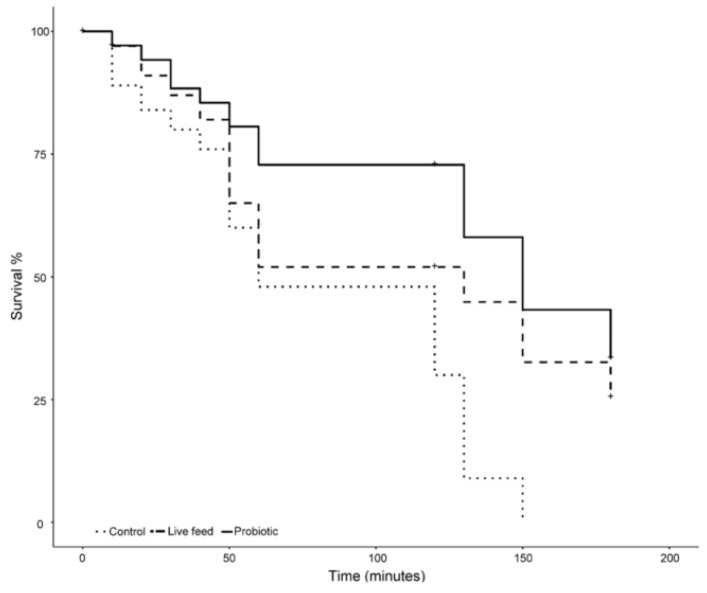
Salinity stress test mortality over a 3 h time frame (*n* = 100), after 21 dph.

**Table 1 microorganisms-08-00238-t001:** Experiment husbandry schedule. Amount of daily feed offered, and recirculation flow changes with time are shown.

DPH	Daily Feed:Rot-Art/mL	Flow (ml/min)
3	10-0	100
4	10-0	100
5	10-0	100
6	10-0	100
7	10-0	100
8	14-0	160
9	14-0	160
10	14-0	160
11	14-0	160
12	14-2	200
13	10-3	200
14	8-4	200
15	0-7	250
16	0-7	250
17	0-8	250
18	0-8	250
19	0-8	250
20	0-8	250
21	End of Trial	250

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
