# Peer review of "Use of Lactic Acid Bacteria During Pikeperch (Sander lucioperca) Larval Rearing"

_microorganisms, 2020, doi:10.3390/microorganisms8020238_

Round 1

Reviewer 1 Report

This work presents the results of a well designed experiment to test the potentially beneficial effects of feeding rotifers and artemia lactobacillus, or by additionally adding lactobacillus into the rearing water.

While the effects on survival and on stress resistance are clear and convincing, I have some reluctance to accept the effects on growth and myomere size as significant. The statistics here are not so clear.

More specific comments are

Line 19: should be "significantly higher" Lines 67-68: Is this sentence as intended. It seems contradictory to me " It is also important that defense mechanisms be present in the immune system before it is fully developed, so that it can proliferate pathogenic bacteria and inhibit colonization Line 178, Figure 1: It seems difficult to believe that Total Length for probiotic is higher at 21dph, in view of the very large standard error displayed. The fact that no significant differences is seen at 16dph is an indication that the 12 and 21dph data are also not really different.The legend says that dots show mean values, is this a mistake, or why would there be a different number of dots for different samples? In the legend, it says " Statistically significant differences between treatments from are marked with an asterisk. : delete "from" Lines 185-189, Fig. 2: similar comments also apply to Fig. 2, here the "maybe significant" differences are seen at 16dph. In addition, the authors somewhat "tweek" the other results as "Although not significant, 12 and 21 dph larvae from the probiotic treatment had a larger myomere": if they are not significant, well, they are NOT. Same comment for the legend. Lines 200-204, Fig. 4: here the effect of "Probiotic" is convincing relative to the two other samples, however the difference between Control and "Life feed" should be qualified as non-significant (not mentioning the 1.1 fold). In the legend " Statistically significant differences between data from 21 dph are marked with an asterisk." Should be " Statistically significant differences between samples are marked with an asterisk. Figure 5: OK, only the legend needs to be corrected: there are no dots and no asterixes. Lines 239-242: I have some problems with the statements "a positive influence of probiotics was observed when looking at other parameters such as total length and myomere height" or " growth was improved". As explained above, this is somewhat over-interpreting the data. Line 259: Is "reducing synthesis of muscle protein" a benefit, similar to reducing cortisol levels? I don't think so

Author Response

Dear Reviewer,

Thank you for your feedback and corrections and helping to improve this manuscript below I try to addressed all your questions and queries, which are also already corrected in the manuscript (highlighted in red)

Concerning the improvement of the English I have sent the manuscript for further improvements to a professional “native speaker” company specialized on scientific papers.

Thank you for spotting my mistake in line 67,  I have corrected accordingly.

About the Figures interpretation I totally I understand the confusion since the figure legend has an incorrect description, please see correct description below (also added to the manuscript)

Dots shown are the out layers, whiskers indicate the maximum and minimum values excluding out layers; the line in the middle of box is the Median value and upper and lower quartiles are the ends of the box. Statistically significant differences between treatments are marked with an asterisk.

With this in mind, figures match statistical analysis and further results. This applies for both e figure 1 and 2.

Concerning the phrase “although significant..”, I agree with you, is not significant (period). I have deleted such sentence.

I have clarified the no significance difference between live feed and control treatment in term of survival figure 4, by stating “no significant” in the text.

I have also corrected the mistakes in the legends as suggested.

After corrected the figure legend explanation about the box plots, I hope my interpretation of the results: “a positive influence of probiotics was….”Is justified.

I have corrected the sentenced about reducing synthesis of muscle protein, is more the opposite effect, I have deleted either way and leave at the reduction of cortisol levels as referenced.

Thank you

Carlos

Reviewer 2 Report

Comments to the Authors of manuscript number: microorganisms-713539 entitled “Use of lactic acid bacteria during pikeperch (Sander lucioperca) larval rearing”.

In the present paper I reviewed Authors present the effect of dietary supplementation of lactic acid bacteria in trails on the larval survival in pikeperch.

The study is very interesting, the manuscript is read with pleasure, however it need small corection.

These three treatment should be described in the manner which allows to distinguish the second treatment from the third treatment. It is currently difficult to understand how these two treatments differ.

L 142 L 145 – ppt- what is whis?

Table 1 – DHP- what is whis?

L 252 – igf - it should be written with a capital letter.

Author Response

Dear Reviewer 2,  Thank you for your feedback and corrections and helping to improve this manuscript below I try to addressed all your questions and queries, which are also already corrected in the
manuscript (highlighted in red)

First: I have tried to clarified the treatmnet description, hopefully is easier to understand now: "The second treatment (live feed) used the same live-feed protocol with the addition of the
probiotic (Bactocell Aqua 100, Pediococcus acidilactici, 1.1011 CFU/g of product, Lallemand
SAS, Blagnac, France) at a daily dose of 1g/m3 in the rotifer and artemia culture tanks giving a
probiotic concentration of 1.105 CFU/mL in the live-feed culture water. In this treatment,
probiotic was only used on the larval feed (rotifer and artemia), therefore, larvae had no direct
external contact with the probiotic, and they only had contact with the probiotics when
ingesting the prey. The third treatment (probiotic) followed the same live-feeding protocol as
the control treatment with rotifers and artemia being exposed to a daily dose 1g/m3 of Bactocell
Aqua 100 during their culture. Additionally, the probiotic product was added daily to the
larval rearing water at a dose of 0.1g/m3 daily over the trial’s duration. In this treatment larvae
had direct external contact with probiotics (in the water) as well as by ingesting the preys".
ppt refers to "parts per thousand" also use as miligrams of salt per liter of freshwater. I have
described the definition on line 20 (first time it appears)

Thank you for spotting the typo in the table 1, is meant to be DPH (days post hatching). I
have corrected the typo, as well as IGF.
Thank you